# Preclinical Evidence of STAT3 Inhibitor Pacritinib Overcoming Temozolomide Resistance via Downregulating miR-21-Enriched Exosomes from M2 Glioblastoma-Associated Macrophages

**DOI:** 10.3390/jcm8070959

**Published:** 2019-07-02

**Authors:** Hao-Yu Chuang, Yu-kai Su, Heng-Wei Liu, Chao-Hsuan Chen, Shao-Chih Chiu, Der-Yang Cho, Shinn-Zong Lin, Yueh-Sheng Chen, Chien-Min Lin

**Affiliations:** 1Graduate Institute of Clinical Medical Science, China Medical University, Taichung 404, Taiwan; 2Department of Neurosurgery, An Nan Hospital, China Medical University, Tainan 709, Taiwan; 3Department of Neurosurgery, China Medical University Beigang Hospital, Yunlin 651, Taiwan; 4Graduate Institute of Clinical Medicine, College of Medicine, Taipei Medical University, Taipei City 11031, Taiwan; 5Department of Neurology, School of Medicine, College of Medicine, Taipei Medical University, Taipei City 11031, Taiwan; 6Division of Neurosurgery, Department of Surgery, Taipei Medical University-Shuang Ho Hospital, New Taipei City 23561, Taiwan; 7Taipei Neuroscience Institute, Taipei Medical University, Taipei 11031, Taiwan; 8Center for Cell Therapy, China Medical University Hospital, Taichung 404, Taiwan; 9Drug Development Center, China Medical University, Taichung 404, Taiwan; 10Graduate Institute of Biomedical Sciences, China Medical University, Taichung 404, Taiwan; 11Department of Neurosurgery, China Medical University Hospital, Taichung 404, Taiwan; 12Bioinnovation Center, Buddhist Tzu Chi Medical Foundation, Hualien 97004, Taiwan; 13Department of Neurosurgery, Tzu Chi University, Hualien Tzu Chi Hospital, Buddhist Tzu Chi Medical Foundation, Hualien 97004, Taiwan; 14Department of Biomedical Imaging and Radiological Science, China Medical University, Taichung 404, Taiwan

**Keywords:** tumor microenvironment (TME), glioblastoma multiforme (GBM), GBM-associated macrophages (GAMs), exosomes, oncomiR-21, STAT3 inhibitor

## Abstract

Background: The tumor microenvironment (TME) plays a crucial role in virtually every aspect of tumorigenesis of glioblastoma multiforme (GBM). A dysfunctional TME promotes drug resistance, disease recurrence, and distant metastasis. Recent evidence indicates that exosomes released by stromal cells within the TME may promote oncogenic phenotypes via transferring signaling molecules such as cytokines, proteins, and microRNAs. Results: In this study, clinical GBM samples were collected and analyzed. We found that GBM-associated macrophages (GAMs) secreted exosomes which were enriched with oncomiR-21. Coculture of GAMs (and GAM-derived exosomes) and GBM cell lines increased GBM cells’ resistance against temozolomide (TMZ) by upregulating the prosurvival gene programmed cell death protein 4 (PDCD4) and stemness markers SRY (sex determining region y)-box 2 (Sox2), signal transducer and activator of transcription 3 (STAT3), Nestin, and miR-21-5p and increasing the M2 cytokines interleukin 6 (IL-6) and transforming growth factor beta 1(TGF-β1) secreted by GBM cells, promoting the M2 polarization of GAMs. Subsequently, pacritinib treatment suppressed GBM tumorigenesis and stemness; more importantly, pacritinib-treated GBM cells showed a markedly reduced ability to secret M2 cytokines and reduced miR-21-enriched exosomes secreted by GAMs. Pacritinib-mediated effects were accompanied by a reduction of oncomiR miR-21-5p, by which the tumor suppressor PDCD4 was targeted. We subsequently established patient-derived xenograft (PDX) models where mice bore patient GBM and GAMs. Treatment with pacritinib and the combination of pacritinib and TMZ appeared to significantly reduce the tumorigenesis of GBM/GAM PDX mice as well as overcome TMZ resistance and M2 polarization of GAMs. Conclusion: In summation, we showed the potential of pacritinib alone or in combination with TMZ to suppress GBM tumorigenesis via modulating STAT3/miR-21/PDCD4 signaling. Further investigations are warranted for adopting pacritinib for the treatment of TMZ-resistant GBM in clinical settings.

## 1. Introduction

Glioblastoma multiforme (GBM) is the most aggressive brain tumor of glial origin and has a poor median survival of 14 months [1]. One of the reasons for its malignancy and challenging therapeutics development lies in its heterogeneous nature at the cellular and molecular levels. It is now generally recognized that GBM is composed of a subpopulation of glioma stem cells (GSCs), capable of tumor initiation and progressive self-renewal upon treatments, and other cells within the tumor microenvironment (TME). The TME contains cancerous cells surrounded by parenchymal cells, including endothelial/vascular cells, microglia, and immune cells [2]. One of the major cell types from the GBM TME is glioblastoma-associated macrophages (GAMs). GAMs have been shown to contribute to the progression of GBM. For instance, the presence of M2 GAMs has been shown to promote the growth and metastasis of GBM cells [2,3]. 

More importantly, emerging evidence indicates the dynamic intercellular communications within the GBM TME via secretions of cytokines, chemicals, and signaling molecules. Among these, secreted exosomes represent a class of small bilayered particles (ranging from 50 to 150 nm in diameter) which have been extensively explored for their roles in GBM tumorigenesis over the past few years [4]. Recent studies have shed light on the diverse functions of exosomes involved in GBM tumorigenesis. For instance, exosomes released from human GBM cell lines contain various types of heat shock proteins and transforming growth factor beta 1(TGF-β1) which are proposed to exert immune suppressive roles in GBM [5]. In addition, serum-derived exosomes from GBM patients and Cerebrospinal fluid (CSF) derived exosomes were shown to contain a high level of miR-221, serving as a potential GBM biomarker [6]. A recent study demonstrated that microglia also communicate and affect the function of glioma via the release of exosomes [7]. These findings suggest that there is a potential area for therapeutics development via interrupting the intracellular communications between GBM cells and their TME by means of exosomes. However, the role of exosomes derived from M2 GAMs has not been fully appreciated. 

In this study, we first demonstrated that when human GBM cell lines were cocultured with clinically isolated glioblastoma-associated macrophages, this significantly enhanced colony formation ability and tumor sphere generation in association with an increased expression of Sox2, STAT3, interleukin 6 (IL-6), and Nestin and a decrease in glial fibrillary acidic protein (GFAP). Subsequently, exosomes released into the culture medium of GAMs were isolated and cocultured with GBM cell lines. A similarly increased tumorigenic property was observed in addition to the increased resistance against temozolomide (TMZ). More importantly, miR-21, a oncomiR, was identified as the most abundant microRNA species in the exosomes released from the GAMs. We then provided evidence for the positive association between miR-21 level and GBM malignancy. Exogenously increased miR-21 in GBM cells increased their ability to polarize GAMs towards the M2 phenotype, and the reduction of miR-21 reversed these properties. In addition, we showed that miR-21-mediated oncogenic properties were associated with their targeting/inhibitory function on PDCD4 (a tumor suppressor). An increased miR-21 level in the GBM cells led to their increased ability to polarize GAMs towards the M2 phenotype by the increased secretion of the M2 cytokines IL-6 and TGF-β1. 

Subsequently, we examined the feasibility of applying pacritinib, an inhibitor of the STAT3-associated pathway, as an anti-GBM agent. We showed that pacritinib treatment significantly reduced cell viability and colony/tumor sphere formation in association with reduced levels of STAT3, Sox2, PDCD4, and miR-21; it also reduced the ability to generate M2 GAMs. Notably, pacritinib-treated GAMs released fewer miR-21-enriched exosomes. Finally, we demonstrated preclinical support for using pacritinib to overcome TMZ-resistance using a TMZ-resistant LN18-bearing mouse model. 

## 2. Materials and Methods

### 2.1. Sample Collection and Cell Culture

Tumor sample and stromal GAMs were collected from our Department of Neurosurgery, Taipei Medical University-Shuang Ho Hospital, under strict adherence to Institutional Review Board (IRB) guidelines (approval numbers: IRB: N201801070 and N201602060). Patients were fully informed and a written consent form was signed prior to the operation. The pathological examination was performed by the Department of Pathology and all verified cases met the criteria of GBM. Samples (tumor samples and stromal cells) were isolated and cultured according to previously established protocols [8,9]. Human GBM cell lines U87MG and LN18 were obtained from the American Type Culture Collection (ATCC) and maintained in DMEM supplemented with 2 mM glutamine, 100 U/mL penicillin, 100 μg/mL streptomycin, and 10% FBS. Neurospheres from both cell lines and clinical samples were generated using tumor-sphere-forming medium containing growth factors supplemented with DMEM-F12 1:1 medium, as previously described [10]. For coculture experiments, a previously established protocol was followed with minor modifications [11]. In brief, U87MG and LN18 (2 × 10^5^ cells) were seeded in a transwell insert (0.4 μm pore size) with GAMs (2.5 × 10^5^ cells) seeded in the lower chamber of a six-well system. Cells were cultured in DMEM medium as described above. Cells were maintained for 48 h and harvested for further analyses. In the case of the exosome coculture, GBM cells were cultured in serum-free DMEM in the presence of exosomes for 48 h and harvested. 

### 2.2. Transfection

In order to explore the functional roles of miR-21 in GBM cells, the upregulation or downregulation of miR-21 was achieved using mimic and inhibitor, respectively. MiR-21-5p mimic (HMI0372, Sigma, St. Louis, MO, USA) and inhibitor (HSTUD0371, Sigma, St. Louis, MO, USA) were transfected into GBM cells using Lipofectamine 2000 reagent (Invitrogen, USA) according to the vendor’s instructions. The change in the expression of miR-21-5p was then determined by real-time PCR (RT-PCR) 48 h post-transfection in both GBM cell lines. hsa-miR-21-5p primers (MPH02337, Abm, Richmond, BC, Canada) were purchased and used for qPCR experiments. 

### 2.3. Exosome Isolation

GAMs were cultured in serum-free medium for 48 h (with and without pacritinib treatment, 0.5 µM) before exosome isolation. Culture medium was collected and a standard procedure was performed accordingly [12]. In short, we carried out a serial centrifugation procedure (500× *g* for 10 min, 1200× *g* for 20 min, and 10,000× *g* for 30 min), followed by filtration with a 0.22 μm pore syringe and a spin at 100,000× *g* for 60 min. The collected pellet was washed in PBS three times before another ultracentrifugation at 100,000× *g* for 60 min. The exosomes were used for further analyses. A small portion of the pellet was processed for transmission electron microscopic examination. In brief, purified exosomes were fixed with 1% glutaraldehyde (1 h, room temperature) and washed, followed by 1% reduced osmium tetroxide fixation (1 h). The sample was washed, stained with 0.3% thiocarbohydrazide, and fixed again in OsO4. Finally, the sample was embedded into Epon. Ultrathin sections were placed on formvar-coated grids. Electron microscopy (EM) analysis was performed as previously described [13]. The flowchart of GBM cell lines either treated with exosomes or mimics or inhibitors is listed in the Appendix A.

### 2.4. miRNA PCR Array Analysis

Total RNA (200 ng) isolated from exosomes derived from GAMs was transcribed to cDNA using the miScript II RT kit (Qiagen, Valencia, CA, USA) according to the protocol provided by the vendor. The miRNA PCR array (Qiagen, Valencia, CA, USA) was used for profiling according to the instructions provided. 

### 2.5. Real-Time PCR

Total RNAs were extracted, purified, and reverse transcribed using the RNeasy kit (Qiagen, Valencia, CA, USA) and OneStep RT-PCR Kit (Qiagen, Valencia, CA, USA). RT-PCR was performed using an I-Cycler IQ Multicolor RT-PCR Detection System (Bio-Rad) with SsoFast Eva Green Supermix (Bio-Rad). All experimental Ct values were normalized against the Ct value of internal control, GAPDH. Relative abundance was determined by 2-ΔΔCt and expressed as fold changes. Primer sequences are listed in Appendix A.

### 2.6. SDS-PAGE and Western Blotting

A standard SDS-PAGE and Western blotting was carried out according to previously established protocols [14]. The primary antibodies used in this study were all purchased from AbCam (Taipei, Taiwan) unless otherwise specified: anti-STAT3 (ab119352, 1:1500); anti-IL-6 (ab6672, 1:500); anti-Sox2 (ab93689, 1:800); anti-Nestin (ab105389, 1:800); anti-CD9 (ab92726, 1:400); anti-CD63 (ab217345, 1:400); anti-CD81 (ab79559. 1:400); anti-actin (ab179467, 1:2000); and anti-tubulin (ab6046, 1:1000). 

### 2.7. In Vivo Xenograft Model

A tumor sample from a GBM patient with TMZ resistance was used to establish the TMZ-resistant mouse model for in vivo evaluation according to previously established protocols [15]. In brief, NOD/SCID mice were anaesthetized (10 mg/kg, ketamine/xylazine and buprenorphine, 0.05 mg/kg, before and after injection). TMZ-resistant LN18 GBM cells (5 × 10^5^ cells) were stereotactically injected into the right striata of the mice. One week postinjection, the mice were randomly divided into vehicle, pacritinib (100 mg/kg, five times/week), TMZ (30 mg/kg, five times/week), or the combination of pacritinib (100 mg/kg) and TMZ (30 mg/kg) groups. Both drugs were administered via oral gavage. Mice were humanely sacrificed by sodium pentobarbital at the end of the experiments. The tumor presence and size were determined in the mice via necropsy and cranial dissection. Tumor samples were harvested for further analysis. The tumor size (average area) was determined from cross sections of the tumor samples. Image J software was used for calculating the tumor size. The animal study protocol was approved by the Animal Care and User Committee at Taipei Medical University (Affidavit of Approval of Animal Use Protocol# Taipei Medical University—LAC-2017-0512).

### 2.8. Statistical Analysis

The miRNA expression levels from the array experiments were analyzed by SDS software version 2.2.2 (Applied Biosystems, foster city, CA, USA). The delat Ct values were calculated against U6 internal control. Heatmaps of differentially expressed miRNAs were analyzed by R software. Other data were analyzed using Student’s *t*-test to determine statistical significance among the different groups. *p*-values (represented by asterisks), where * *p* < 0.05; ** *p* < 0.01; *** *p* < 0.001; **** *p* < 0.0001.

## 3. Results

### 3.1. M2 Polarization of GAMs Promotes GBM Tumorigenesis

Initially, we cocultured clinically isolated GAMs with human GBM cell lines U87MG and LN18, which showed increased colony (Figure 1A) and neurosphere (Figure 1B) forming abilities. Consistently, qPCR analysis demonstrated that the presence of GAMs was associated with an increased mRNA level of stemness markers (Figure 1C). The results from the Western blots were consistent where the expression of Sox2, Oct4, Wnt, and Nestin were elevated, while GFAP was decreased in the presence of GAMs (Figure 1D). 

### 3.2. Exosome Enriched with miR-21 from GAMs Promotes Tumorigenic Properties

We further investigated the underlying tumorigenesis by isolating and characterizing exosomes secreted into the culture medium by GAMs. First, we used different markers for exosomes—CD9, CD63, and CD81—to verify the identity of the exosomes isolated from the GAMs (Figure 2A). Next, we showed that incubation of GAM-derived exosomes significantly increased TMZ resistance in both U87MG and LN18 cells (Figure 2B). For example, the estimated IC_50_ value for U87MG increased approximately 4-fold after incubation with GAM-derived exosomes, while this was even more significant in the LN18 cells after exosome treatment. This increased TMZ resistance was accompanied by increased colony-forming (Figure 2C) and tumor-sphere-forming (Figure 2D) abilities. We then screened a small cohort of microRNAs in two batches of GAM-derived exosomes and found that miR-21 appeared to be the most abundant microRNA (Figure 2E). As shown in the heatmap, the miR-21 level appeared to be the most enriched in the exosomes collected from two samples of GAMs. 

### 3.3. MiR-21 Is Associated with GBM Tumorigenic Properties

Next, we examined the effects of miR-21 on GBM cells by gene silencing and overexpression techniques. First, we demonstrated that miR-21-5p level increased in both U87MG and LN18 cells after being cocultured with GAM-derived exosomes (Figure 3A). We then transfected GBM (postincubation with GAM exosomes) with either mimic or inhibitor molecules of miR-21-5p. We found that the stemness markers Sox2, Oct4, Wnt, STAT3, and Nestin were all significantly increased when mimic miR-21-5p was added to both cells, while the opposite occurred after the miR-21-5p level was inhibited (Figure 3B). The Western blotting results agreed with the real-time PCR results (Figure 3C), where an increased miR-21-5p level led to the increased expression of Sox2, Oct4, STAT3, Akt, Nestin, and Wnt and a decreased level of GFAP. More importantly, tumorigenic properties such as colony formation and tumor sphere formation were also positively correlated with the level of miR-21-5p. For instance, an increased miR-21-5p level by mimic molecules led to an increased number of colonies (Figure 3D) and neurospheres (Figure 3E) generated, and the opposite was true with a decreased level of miR-21-5p with the inhibitor treatment. Furthermore, miR-21-5p mimic transfection made both U87MG and LN18 cells more resistant against TMZ, whereas miR-21-5p inhibitor reversed the resistance (Figure 3F). 

### 3.4. STAT3 and PDCD4 are Targets of miR-21-5p

Subsequently, we examined the potential target(s) for miR-21-5p using bioinformatics tools and we identified STAT3, a well-known oncogene, and PDCD4, an established tumor suppressor, as the top-ranking candidates from all three algorithms used (PITA, miRmap, and miRanda). A potential site of interaction between miR-21-5p and STAT3 and PDCD4 was identified in the 3’UTR (upper panel, Figure 4A); more importantly, based on TCGA database, a strong negative correlation between the expression level of PDCD4 and miR-21-5p was established within a cohort of GBM patients (*n* = 525, lower panel, Figure 4B). We then demonstrated that increased miR-21-5p by mimic molecules in both U87MG and LN18 cells supported the negative correlation between the expression of miR-21-5p and PDCD4. Conversely, a decreased level of miR-21-5p by inhibitor molecules restored the expression of PDCD4 (Figure 4C). We then cocultured miR-21-5p-silenced GBM cells with GAMs and observed a significantly reduced M2 signature (CD68+/CD206+) (Figure 4D). More importantly, the M2 cytokines VEGF, TGF-β1, and IL-6 released by GAMs cocultured with miR-21-5p-silenced U87MG cells were significantly reduced and restored partially after cocultured miR-21-5p-silenced U87MG were transfected with mimic of miR-21-5p (Figure 4E). 

### 3.5. Pacritinib Suppresses GBM Tumorigenesis and M2 Polarization of GAMs

Elevated STAT3 signaling has been attributed to the malignancy of GBM and the generation of glioma stem cells [16]. In addition, increased STAT3 signaling is associated with the increased miR-21 level in the promotion of tumorigenesis [17,18]. Based on these premises, we examined a clinical STAT3 inhibitor, pacritinib, for its potential GBM inhibitory effects. We found that pacritinib treatment significantly suppressed the cell viability of both U87MG and LN18 cells at low IC_50_ values (0.5 and 1.7 µM, respectively) (Figure 5A). Subsequently, we showed that pacritinib-treated U87MG and LN18 cells contained a significantly lower ability to generate M2-polarized GAMs (Figure 5B), as reflected by the reduced CD206 (M2 marker) and increased TNF-α (M1 marker). In addition, the addition of pacritinib prominently suppressed colony formation (Figure 5C) and tumor sphere generation (Figure 5D). Furthermore, pacritinib treatment led to a decreased expression of Sox2, PDCD4, and STAT3; more importantly, the level of miR-21-5p in both GBM cell lines was suppressed as well (Figure 5E). Notably, pacritinib treatment led to significantly reduced exosome release and a corresponding level of miR-21-5p from GAMs (Figure 5F). 

### 3.6. In Vivo Evaluation of Pacritinib

Finally, we evaluated the potential of using pacritinib as a treatment for GBM using a preclinical mouse model bearing TMZ-resistant LN18 cells (cocultured with exosomes isolated from GAMs). Representative brain slices showed that a single treatment of pacritinib suppressed the tumorigenesis of TMZ-resistant LN18 cells compared to TMZ single treatment and vehicle control (Figure 6A). Notably, there was no significant difference in tumor size between vehicle control and TMZ single treatment groups (Figure 6B), while the combination of pacritinib and TMZ appeared to produce the most significant inhibitory effect on tumor progression (right panel, Figure 6B). In support, tumor samples harvested from the combination of pacritinib and TMZ showed the lowest level of STAT3, Sox2, PDCD4, and miR-21-5p and an increased level of GFAP (Figure 6C). Microglial cells isolated from the single pacritinib treatment and the combination of pacritinib and TMZ groups also demonstrated a significantly reduced CD206 mRNA level and an increased TNF-α level (Figure 6D). The overall median survival was significantly increased in each treatment group compared with vehicle control (Figure 6E). Median survival was 19 days for vehicle control, 24 days for TMZ (*p* = 0.024, compared with control ), 26.5 days for pacritinib (*p* = 0.0098, compared with control ), and 32.5 day for combination of pacritinib and TMZ (*p* = 0.0006, compared with control, *p* = 0.0092, compared with TMZ, *p* = 0.0219, compared with pacritinib) (Figure 6F).

## 4. Discussion

Despite advances in therapeutics development over the past decade, GBM remains challenging to treat due to its heterogeneity and malignant nature. The tumor microenvironment plays a crucial role in promoting GBM tumorigenesis. GAMs have been shown to be one of the key players in the GBM microenvironment. We first demonstrated that clinical samples of GAMs promoted GBM tumorigenesis. For instance, U87MG and LN18 GBM cells cocultured with clinical M2 GAMs showed increased colony-forming and tumor-sphere-generating abilities in association with increased stemness markers Sox2, STAT3, Wnt, and Nestin in the GBM cells. Accumulating evidence has supported the observations where GAMs induced epithelial–mesenchymal transition (EMT) in GBM cells and subsequently generated properties of GSCs [19]. In addition, our observations were in agreement with previous studies, where interactions between GBM and GAMs increased CD133+ GSCs and malignant phenotypes [20,21]. GAM-mediated GBM-promoting effects were through different communicating molecules such as M2 cytokines (IL-6, VEGF, and TGF-β1) [2]. Here, we showed that the presence of GAMs promoted GBM tumorigenesis and stemness not only via the cytokines but also through the aid of exosomes. More specifically, we found that GBM cells incubated with exosomes derived from GAMs exhibited enhanced ability in colony and tumor sphere formation; more importantly, exosome-incubated GBM cells became more resistant against TMZ. Emerging evidence indicates the functional roles of exosomes in GBM tumorigenesis. A recent study showed that exosomes secreted from GBM cells promoted the oncogenic transformation of astrocytes in the tumor microenvironment [22]. This observation complements the results of our study, which demonstrated intimate communication between the tumor microenvironment and tumor cells via the exchange of exosomes. 

We performed an array analysis on the exosomes secreted by GAMs and found that the most abundant microRNA species was miR-21. Notably, a recent review points out that miR-21 plays a pivotal role in GBM pathogenesis, where miR-21 functions through the modulation of the insulin-like-growth-factor-associated signaling pathway, RECK, and TIMP3 to promote GBM tumorigenesis [23]. Our results provided an added feature of miR-21 in GBM tumorigenesis, where miR-21 was enriched in the exosomes secreted by GAMs. It is very plausible that GAM-derived miR-21-enriched exosomes were incorporated into GBM cells and executed their tumor-promoting functions. It has been well demonstrated that the transfer and uptake of exosomes between donor and recipient cells represents one of the major routes for intercellular communications in many diseases, including cancer [24]. We provided support that increased miR-21-5p in GBM cells by miR-21-5p mimic molecules resulted in similar tumorigenic and stemness properties in GBM cells cocultured with GAM-derived exosomes; the reduction of miR-21-5p significantly reduced the tumorigenic properties in both GBM cell lines. Furthermore, GBM cells transfected with miR-21-5p inhibitor showed a significantly reduced ability to generate M2 GAMs, based on our coculture experiments; this was attributed to the decreased secretion of M2 cytokines such as IL-6 and VEGF by miR-21-5p-silenced GBM cells and an increased secretion of TNF-α, an M1 marker. More importantly, we provided evidence that miR-21-5p targets PDCD4, a tumor suppressor in both GBM cell lines. PDCD4 has been shown to be frequently suppressed in GBM cells and is associated with poor prognosis [25,26]. In agreement with our results, a previous study also demonstrated that PDCD4 was targeted by miR-21 in GBM [27].

According to our experimental results, miR-416a ranks as the second-most abundant microRNA species in the GAM-secreted exosomes. It has been shown that miR-416a plays a key role in the progression of malignant melanoma via the activation of notch signaling [28]. The activation of notch signaling has also been shown to be responsible for the generation of GSCs [29,30]. The fact that miR-21 and miR-416a, two powerful oncogenic microRNA molecules, were enriched in the GAM exosomes further supports our notion that GAMs play a key contributing role in GBM malignancy and should be targeted in treatment design. Currently, the role of exosomal miR-416a in GBM tumorigenesis is under intense investigation in our laboratory. 

Since targeting microRNA for therapeutic purposes still remains challenging, miR-21-5p represents a potential therapeutic target. Thus, we evaluated the feasibility of using a small-molecule agent which may indirectly increase the miR-21 level to convey therapeutic functions in GBM. STAT3 signaling has been shown to be important in GBM tumorigenesis as well as linked to the expression of miR-21 [17,31,32]. Based on these premises, we evaluated pacritinib, a recent FDA-approved inhibitor of STAT3/JAK2 signaling for treating myelofibrosis [33,34]. We found pacritinib treatment suppressed cell viability and colony/tumor sphere formation in association with decreased expression of STAT3, Sox2, PDCD4, and miR-21-5p and increased GFAP expression. Equally important, GAMs cocultured with pacritinib-treated U87MG and LN18 GBM cells showed a significantly reduced M2 marker (CD206) and increased M1 marker (TNF-α), strongly suggesting pacritinib not only suppressed GBM tumorigenesis but also affected GAM polarization. These tumor inhibitory and tumor microenvironment normalizing effects of pacritinib could be attributed to the suppression of STAT3/JAK2 signaling. Our observations were supported by a recent report that the inhibition of the JAK/STAT3 pathway resulted in disrupted intercellular communications between microglia and GBM cells [35] and pronounced anti-GBM effects [36,37]. In addition, we found that pacritinib treatment was able to suppress the number of miR-21-enriched exosomes secreted by GAMs. 

Finally, we provided support for combining pacritinib with TMZ using a TMZ-resistant GBM mouse model. A single treatment of pacritinib was sufficient to suppress GBM growth, while the combination of pacritinib and TMZ exerted the most significant inhibitory effect. Several studies have demonstrated the benefit of using a STAT3 inhibitor to overcome TMZ resistance [38,39]. Notably, one report showed that STAT3 inhibitor treatment promoted the infiltration of tumoricidal lymphocytes [40]. Another study also lends support to our results, where the sequential combination of STAT3 inhibition and TMZ led to the induction of GBM apoptosis with an increased level of miR-21 [41]. This is consistent with another previous study that combined treatment with pacritinib and TMZ to dramatically reduce the activity of the JAK2/STAT3 pathway. This highlights the potential for pacritinib to be a useful adjuvant therapy with the standard-of-care TMZ. Additionally, pacritinib could be used as a salvage therapy for patients with a TMZ-resistant recurrent disease, as STAT3 inhibition sensitizes TMZ-resistant, patient-derived brain-tumor-initiating cell (BTIC) cultures [42].

## 5. Conclusions

In conclusion, as shown in the scheme in Figure 7, we have provided translational evidence that miR-21-enriched GAM-derived exosomes contribute to GBM malignancy via increasing stemness. The feasibility of using pacritinib to modulate STAT3/miR-21/PDCD4 signaling was demonstrated using both in vitro and in vivo GBM models. Further investigation is warranted for conducting potential clinical trials for GBM patients experiencing TMZ resistance.

## 6. Ethics Approval and Consent to Participate

Clinical samples were collected from Taipei Medical University (Taipei, Taiwan). All enrolled patients gave written informed consent for their tissues to be used for scientific research. The study was approved by the IRB of the Taipei Medical University (IRB: N201801070 and N201602060), consistent with the recommendations of the Declaration of Helsinki for biomedical research (Taipei Medical University (Taipei, Taiwan) and following standard institutional protocol for human research. Moreover, the animal study protocol was approved by the Animal Care and User Committee at Taipei Medical University (Taipei, Taiwan) (Affidavit of Approval of Animal Use Protocol# Taipei Medical University—LAC-2017-0512).

## Figures and Tables

**Figure 1 jcm-08-00959-f001:**
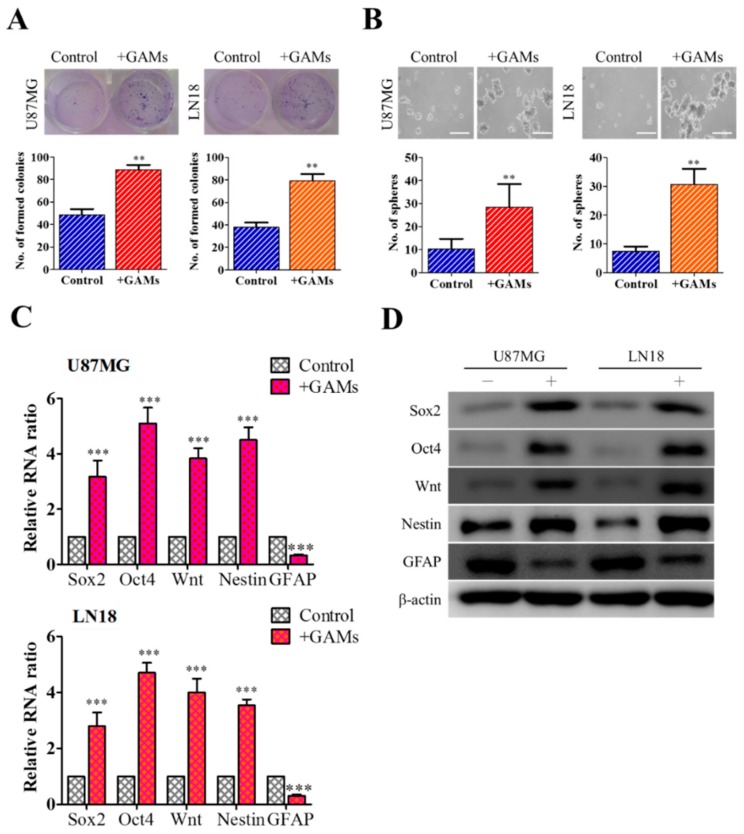
M2 glioblastoma multiforme (GBM)-associated macrophages (GAMs) promote GBM tumorigenesis. GBM cells U87MG and LN18 cocultured with M2 GAMs showed significantly increased colony forming ability (**A**) and tumor sphere generating ability (**B**) as compared to their parental controls. Comparative real-time PCR (**C**) and Western blots (**D**) showed that M2 GAM cocultured GBM cells expressed a significantly higher level of stemness markers, Sox2, Oct4, Wnt, and Nestin while GFAP was reduced. Scale lengths = 100 μm, * *p* < 0.05; ** *p* < 0.01; *** *p* < 0.001.

**Figure 2 jcm-08-00959-f002:**
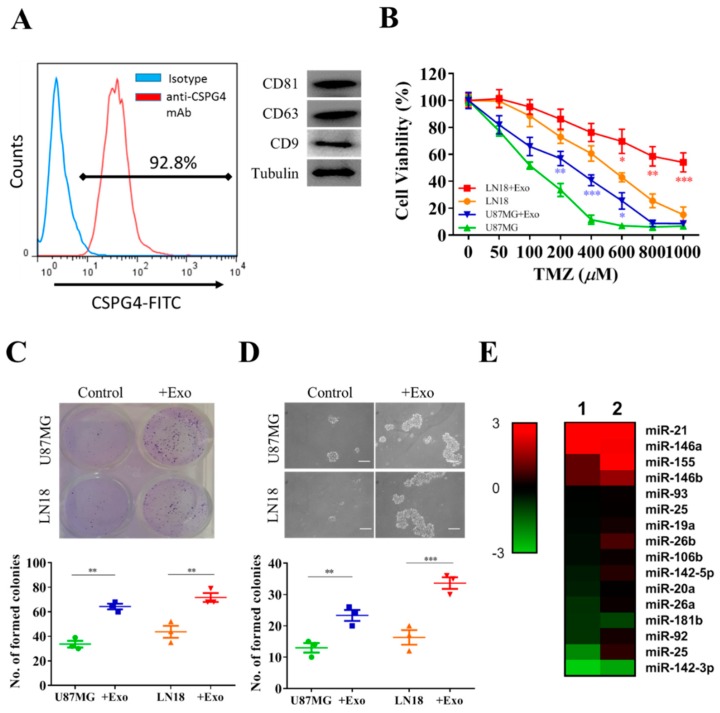
GAM-derived exosomes harbor miR-21, which promote GBM tumorigenesis. (**A**) Representative transmission electronic micrograph of exosomes isolated from clinical GAMs (left); Western blot validation of exosomes isolated from GAM culture medium showed the expression of CD9, CD63, and CD81. (**B**) Increased temozolomide (TMZ) resistance in U87MG and LN18 cells cocultured with exosomes (+exo). Enhanced colony-forming ability (**C**) and tumor-sphere-generating ability (**D**) in the presence GAM-derived exosomes. (**E**) MicroRNA profiling analyses showed that exosomes (two samples) isolated from M2 GAMs contained a high level of miR-21. Scale lengths = 100 μm, * *p* < 0.05; ** *p* < 0.01; *** *p* < 0.001.

**Figure 3 jcm-08-00959-f003:**
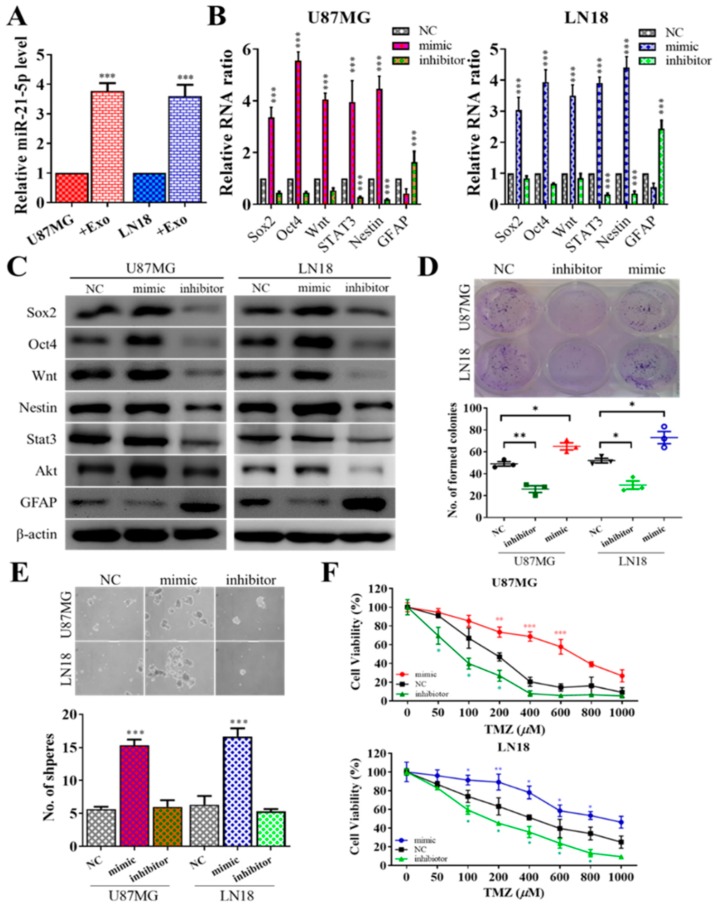
GAM-derived exosomes promoted GBM tumorigenesis via miR-21-5p. (**A**) U87MG and LN18 cells incubated with GAM-derived exosomes showed a significantly increased level of miR-21-5p. GBM tumorigenesis was associated with miR-21-5p. Increased miR-21-5p level (by mimic molecules) in GBM cells showed an increased mRNA level of Sox2, Oct4, Wnt, STAT3, and Nestin or protein level of Sox2, Oct4, STAT3, Akt, Wnt, and Nestin with decreased GFAP, while a decrease in miR-21-5p (inhibitor) led to the opposite phenomenon (**B**,**C**). Incubation with GAM-derived exosomes increased colony-forming ability (**D**) and tumor-sphere-generating ability (**E**) in both U87MG and LN18 cells. (**F**) U87MG and LN18 cells transfected miR-21-5p mimic molecules (increased miR-21-5p level) resulted in significantly increased TMZ resistance, while there was reduced miR-21-59 and decreased TMZ resistance. Scale lengths = 100 μm, * *p* < 0.05; ** *p* < 0.01; *** *p* < 0.001.

**Figure 4 jcm-08-00959-f004:**
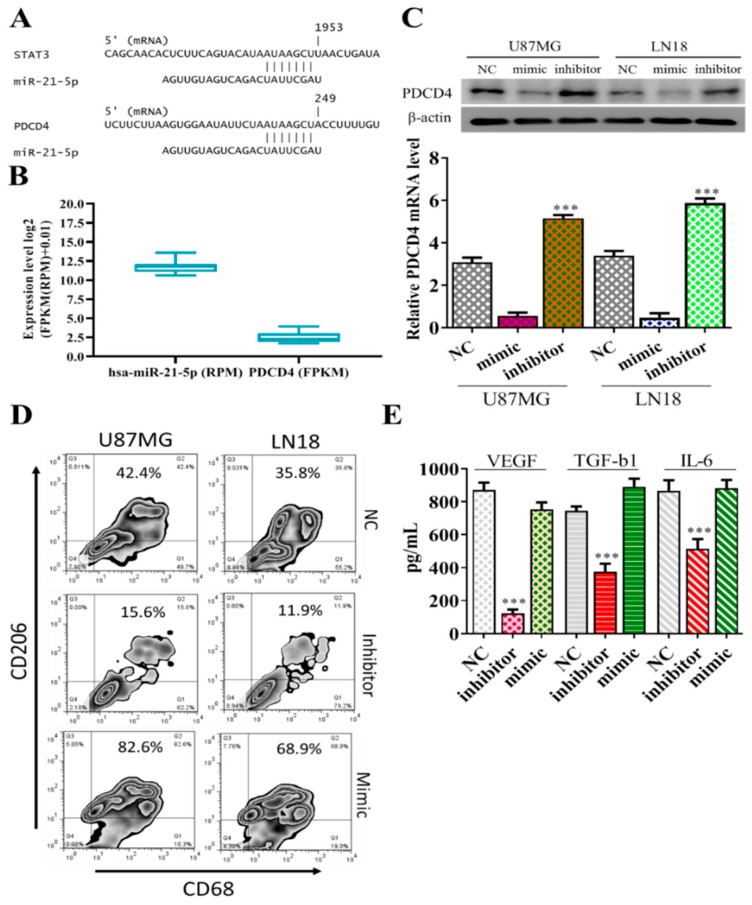
MiR-21-5p targets STAT3 and PDCD4. (**A**) The bioinformatics tool shows miR-21-5p binding to the 3’UTR of STAT3 and PDCD4 (upper panel). (**B**) A negative correlation between the expression of miR-21-5p and PDCD4 in the GBM database (*n* = 525, TCGA). (**C**) An increased miR-21-5p level (by mimic molecule, lane M) led to significantly reduced PDCD4 expression in both U87MG and LN18 cells; the reverse was true with the inhibitor of miR-21-5p. (**D**) Flow cytometry analysis showed a significantly reduced CD206^+^/CD68^+^ population in GAMs cocultured with miR-21-5p-silenced U87MG and LN18 cells; the reverse was observed in miR-21-5p mimic transfected coculture experiments. (**E**) The inhibitor of miR-21-5p resulted in the reduction of VEGF, TGF- β1, and IL-6 secreted by the U87MG cells into the culture medium. *** *p* < 0.001.

**Figure 5 jcm-08-00959-f005:**
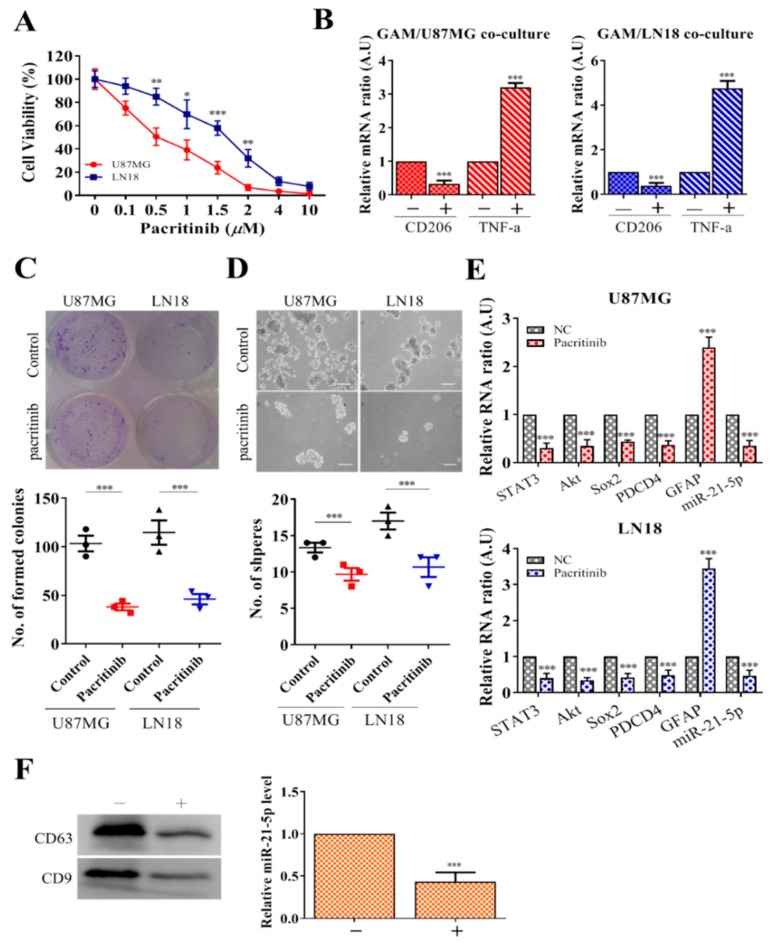
Pacritinib treatment suppresses GBM tumorigenesis and glioma stem cell (GSC) properties. (**A**) Pacritinib treatment significantly suppressed both U87MG and LN18 cells (approximate IC_50_ values 0.5 and 1.5 µM, respectively). (**B**) Pacritinib treatment significantly reduced GBM cells’ ability to induce M2 GAMs. CD206 mRNA in GAMs was significantly reduced, while TNF-α was increased. Pacritinib treatment significantly reduced colony formation (**C**) and tumor sphere generation (**D**) in both U87MG and LN18 cells. (**E**) Pacritinib treatment led to a significantly reduced mRNA level of STAT3, Akt, Sox2, PDCD4, and miR-21-5p and increased GFAP in both U87MG and LN18 cells. (**F**) GAMs treated with pacritinib resulted in the decreased release of exosomes. Western blot of exosomes collected from GAMs showed a significantly lower abundance of exosomes (CD63 and CD9, markers of exosomes). The exosomes collected showed a significantly lower miR-21-5p level. Scale lengths = 100 μm, * *p* < 0.05; ** *p* < 0.01; *** *p* < 0.001.

**Figure 6 jcm-08-00959-f006:**
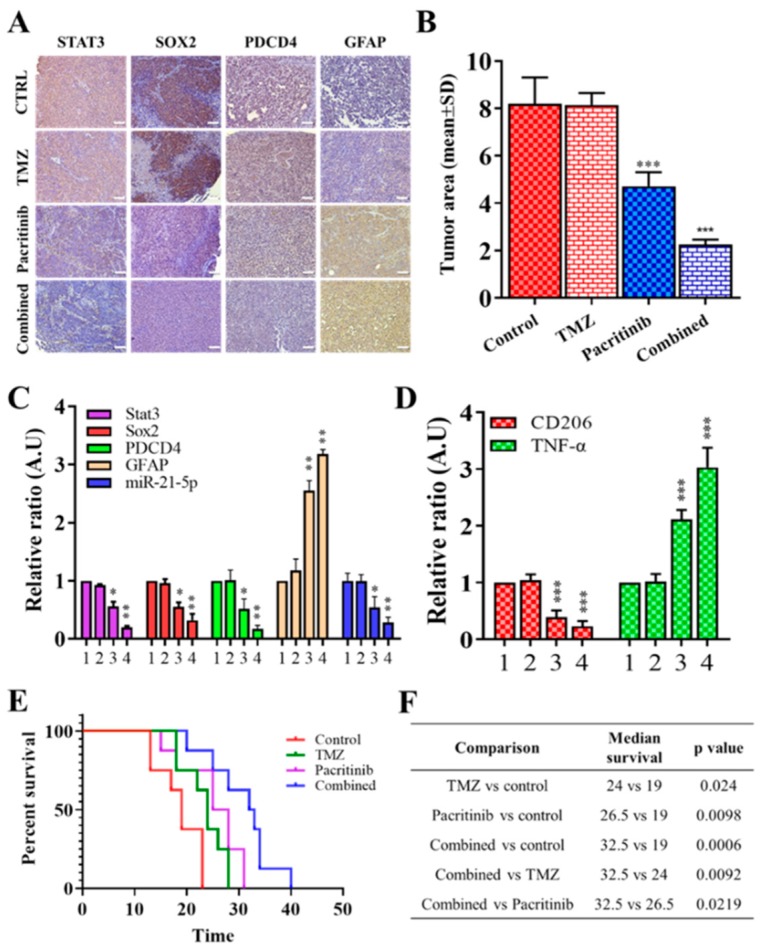
In vivo evaluation of pacritinib for treating GBM and reducing M2 GAMs in TMZ-resistant LN18 bearing mice. (**A**) Immunohistochemical staining in TMZ-resistant LN18-bearing mice showed that treatment in the pacritinib group and pacritinib/TMZ combination group suppressed tumorigenesis. (**B**) The tumor size showed that the significantly reduced tumor size in the pacritinib group and the combination of pacritinib and TMZ group led to the most significantly reduced tumor size. NS, statistically nonsignificant. (**C**) Comparative real-time PCR analyses showed the reduced mRNA level of STAT3, Sox2, PDCD4, and miR-21-5p and the increased GFAP expression in the pacritinib group and pacritinib/TMZ combination group (lane 1, control; lane 2, TMZ alone; lane 3, pacritinib alone; lane 4, pacritinib/TMZ combination). (**D**) M2 GAMs from tumor samples showed a significantly reduced CD206 (M2 marker) mRNA level (lane 3, pacritinib alone; lane 4, pacritinib/TMZ combination) and an increase in TNF-α (lanes 3 and 4). (**E**) Kaplan–Meier survival curve and (**F**) statistical comparisons showed increased median overall survival in TMZ, pacritinib, and pacritinib/TMZ combination groups. Scale lengths = 50 μm, * *p* < 0.05; ** *p* < 0.01; *** *p* < 0.001.

**Figure 7 jcm-08-00959-f007:**
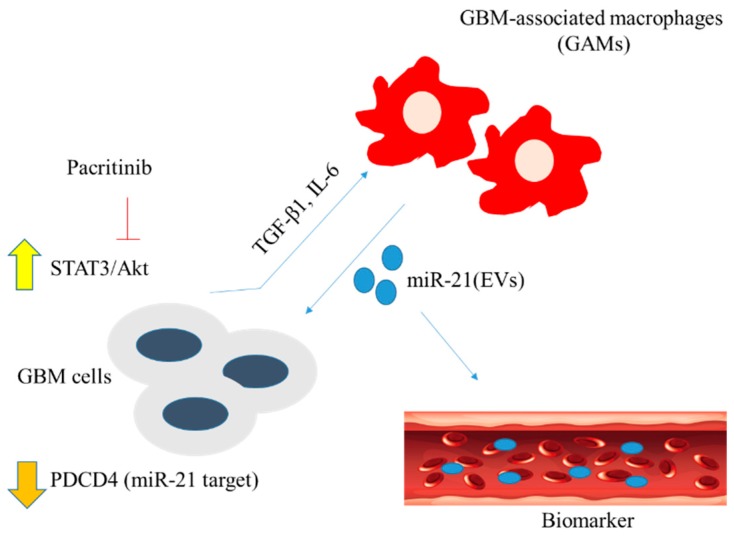
GAMs in the tumor microenvironment promote the survival of GBM cells via miR-21-enriched extracellular microvesicles (EVs). Mir-21 targets and suppresses the expression of tumor suppressor PDCD4 in GBM cells, leading to the elevated STAT3/Akt signaling. In turn, GBM cells secrete inflammatory cytokines TGF-β1 and IL-6 and promote M2 polarization. Pacritinib (STAT3 inhibitor) treatment suppresses GBM tumorigenesis by inhibiting STAT3 signaling and reducing M2 polarization of GAMs.

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
