# Peer review of "Preclinical Evidence of STAT3 Inhibitor Pacritinib Overcoming Temozolomide Resistance via Downregulating miR-21-Enriched Exosomes from M2 Glioblastoma-Associated Macrophages"

_jcm, 2019, doi:10.3390/jcm8070959_

Round 1
Reviewer 1 Report
I would like to congratulate authors for presenting an important piece of research in GBM.
Authors indicate that GBM-derived macrophages secrete exosomes that are enriched in miR-21. Either by treating the GBM established cell line with GAMs or exosomes or mimics they were able to demonstrate an increase in colony formation, spheroid formation, increased TMZ resistance, increased stemness markers expression and priming GAMs to M2 cytokines milieu. The authors also demonstrated that the opposite effect on the GBM cell behaviour is achievable by inhibiting miR-21.
The most significant finding i feel is their ability to demonstrate the clinical potential of Pacrinitib; in increasing TMZ sensitivity in xenografts which precisely is achieved by targeting miR-21 pathway.
Minor points -
1) Please spell/grammar check throughout the text e.g., line 113-114,129, 145, 218
2) Can the authors try and change the heading for Fig2 as it is a bit confusing-
Figure 2 GAM secreted miR-21 enriched exosomes promoted GBM tumorigenesis.
Alternative heading - GAM derived exosomes harbour miR-21 and these promote GBM tumorigenesis.
3) Since so many combination of experiments were performed I urge authors to include a flow chart of experiments / process in supplementary material for ease of understanding of the wider audiences.
(a) GBM cell line + GAM isolated from clinical specimen
(b) GAM cultured > Exosome isolated > co-cultured with GBM cell lines
(c) GBM cell lines either treated with exosomes or mimics or inhibitors etc etc
Author Response
Answers to the comments:
Point-by-point responses to reviewer’s comments:
We would like to thank the reviewer for the thorough reading of our manuscript as well as their valuable comments. We have followed their comments closely and feel that they have further strengthened the manuscript. Below are our point-by-point responses.
Q1: Reviewer #1: I would like to congratulate authors for presenting an important piece of research in GBM. Authors indicate that GBM-derived macrophages secrete exosomes that are enriched in miR-21. Either by treating the GBM established cell line with GAMs or exosomes or mimics they were able to demonstrate an increase in colony formation, spheroid formation, increased TMZ resistance, increased stemness markers expression and priming GAMs to M2 cytokines milieu. The authors also demonstrated that the opposite effect on the GBM cell behaviour is achievable by inhibiting miR-21. The most significant finding i feel is their ability to demonstrate the clinical potential of Pacrinitib; in increasing TMZ sensitivity in xenografts which precisely is achieved by targeting miR-21 pathway.
A1: We thank the reviewer for all the comments and suggestions made. We find them non-prejudicial and helpful. We have revised our manuscript once again based on these comments and do hope we have now addressed all the reviewer’s concerns and now meet the threshold for acceptance.
Q2: Reviewer #1: Minor points. Please spell/grammar check throughout the text e.g., line 113-114,129, 145, 218.
A2: We thank the reviewer for these comments. We correct the spell/grammar check throughout the text e.g., line 113-114,129, 145, 218. Please kindly see our R1 Revised Section, Lines line 113-114, 129, 145, 218.
Q2: Reviewer #1: Can the authors try and change the heading for Fig2 as it is a bit confusing-Figure 2 GAM secreted miR-21 enriched exosomes promoted GBM tumorigenesis. Alternative heading - GAM derived exosomes harbour miR-21 and these promote GBM tumorigenesis.
A2: We thank the reviewer for these comments. We have now corrected these to address the reviewer’s concern. Please kindly see R1 Revised Figure 2 legend, Lines 212-219.
Figure 2 GAM derived exosomes harbor miR-21 and these promote GBM tumorigensis (A) Representative transmission electronic micrograph of exosomes isolated from clinical GAMs (left); western blot validation of exosomes isolated from GAM culture medium showed the expression of CD9, CD63 and CD81. (B) Increased TMZ resistance in U87MG and LN18 cells co-cultured with exosomes (+exo). Enhanced colony forming ability (C) and tumor sphere generating ability (D) in the presence GAM-derived exosomes. (E) MicroRNA profiling analyses showed that exosomes (two samples) isolated from M2 GAMs contained a high level of miR-21. Scale lengths = 100 μm, *p<0.05; **p<0.01; ***p<0.001.< span="">
Q3: Reviewer #1: Since so many combinations of experiments were performed, I urge authors to include a flow chart of experiments / process in supplementary material for ease of understanding of the wider audiences.
A3: We thank the reviewer for this comment. We have included a flow chart of experiments / process in supplementary material for address the reviewer’s concern. Please kindly see our R1 New Supplementary Fig.S6, Fig.S7, and Fig.S8. Please kindly see R1 Revised Supplementary Materials, Lines 429-434.
Supplementary Figure S6. Flow chart of GBM cell line and GAMs from clinical human GBM specimen isolation.
Supplementary Figure S7. Flow chart of Exosome isolation
Supplementary Figure S8. Flow chart of GBM cell lines either treated with exosomes or mimics or inhibitors

Reviewer 2 Report
Chuang and collaborators characterized the exosomes extracted from glioblastoma-associated macrophages, and highlighted the role of STAT3 in témozolomide resistance.
This is an interesting study, with some new findings which can be of interest for neuro-oncology specialists.
I consider that this manuscript is not ready to be published and I propose extra experiments and corrections for improving the quality of the manuscript.
The authors should revise their manuscript for avoiding typing mistakes, missenses. For e.g., what is GAM definition: glioblastoma-associated macrophages (title), glioma associated macrophages (line 62), or glioma associated microglia (line 314) ?
Sentence lines 196-197 is without verb.
Major points:
- Pacritinib was already used in another study: “The JAK2/STAT3 inhibitor pacritinib effectively inhibits patient-derived GBM brain tumor initiating cells in vitro and when used in combination with temozolomide increases survival in an orthotopic xenograft model, PLoS One. 2017; 12(12): e0189670.”. This publication is already describing the role of pacritinib in combined with TMZ. Can the authors precise the gain in mouse survival when treated with both pacritinib and TMZ, in their own study ?
- The authors claim that miR-21-5p binds on STAT3 mRNA but nothing is described on STAT3 after Figure 4A. Can the authors add western blot for STAT3 in Figure 3C ?
- Many statistics were forgotten: Figures 2B, 3D, 3F, 5C, 5D, 6B. Please add them.
- Figure 3: please define NC/M/I in the legend and there is a mistake in 3E as I and M are not on the correct panels (images are extremely blurry).
- The overall quality of the figures is very poor, it is hard to correctly analyze the data and evaluate their quality: Figure 1B, graphs are blurry; Figure 3B, graph legends are impossible to read and the Western blots from 3C are too small; Figure 3E, quality of images too low, 3D is too small; Figure 4B is of very bad quality.
- Legends of the figures are not enough detailed: authors need to precise the scale lengths, the statistical test used for the graphs.
Author Response
Q1: Reviewer #2: Chuang and collaborators characterized the exosomes extracted from glioblastoma-associated macrophages and highlighted the role of STAT3 in temozolomide resistance. This is an interesting study, with some new findings which can be of interest for neuro-oncology specialists. I consider that this manuscript is not ready to be published and I propose extra experiments and corrections for improving the quality of the manuscript.
A1: We thank the reviewer for all the comments and suggestions made. We find them non-prejudicial and helpful. We have revised our manuscript once again based on these comments and do hope we have now addressed all the reviewer’s concerns and now meet the threshold for acceptance.
Q2: Reviewer #2: The authors should revise their manuscript for avoiding typing mistakes, missenses. For e.g., what is GAM definition: glioblastoma-associated macrophages (title), glioma associated macrophages (line 62), or glioma associated microglia (line 314) ?
Sentence lines 196-197 is without verb.
A2: We thank the reviewer for these comments. We have now corrected these to address the reviewer’s concern. Please kindly see our R1 Revised Section, Line 62 and 332.
Please kindly see R1 Revised Figure 1 legend, Lines 192-197.
Figure 1 M2 GAMs promotes GBM tumorigenesis. GBM cells U87MG and LN18 co-cultured with M2 GAMs showed a significantly increased colony forming ability (A) and tumor sphere generating ability (B) as compared to their parental controls. Comparative real-time PCR (C) and western blots (D) showed that M2 GAM co-cultured GBM cells expressed a significantly higher level of stemness markers, Sox2, STAT3, Wnt, and Nestin while GFAP was reduced. Scale lengths = 100 μm, *p<0.05; **p<0.01; ***p<0.001.< span="">
Q3: Reviewer #2: - Pacritinib was already used in another study: “The JAK2/STAT3 inhibitor pacritinib effectively inhibits patient-derived GBM brain tumor initiating cells in vitro and when used in combination with temozolomide increases survival in an orthotopic xenograft model, PLoS One. 2017; 12(12): e0189670.”. This publication is already describing the role of pacritinib in combined with TMZ. Can the authors precise the gain in mouse survival when treated with both pacritinib and TMZ, in their own study?
A3: We thank the reviewer for this important observation. We have now added on the new data of mouse survival when treated with both pacritinib and TMZ in new Figure 6E and Figure 6F
Please kindly see our R1 Revised Results Section, Line 299 and 314.
3.6. In vivo evaluation of pacritinib
Finally, we evaluated the potential of using pacritinib as a treatment for GBM using preclinical mouse model bearing TMZ-resistant LN18 cells (co-cultured with exosomes isolated from GAMs). Representative brain slices showed that single treatment of pancritinib suppressed the tumorigenesis of TMZ-resistant LN18 cells as compared to TMZ single treatment and vehicle control (Fig. 6A). Notably, there was no significant difference in tumor size between vehicle control and TMZ single treatment group (Fig. 6B) while the combination of pacritinib and TMZ appeared to produce the most significant inhibitory effect on tumor progression (right panel, Fig. 6B). In support, tumor samples harvested from the combination of pacritinib and TMZ showed the lowest level of STAT3, Sox2, PDCD4, and miR-21-5p while increased level of GFAP (Fig. 6C). Microglial cells isolated from the single pacritinib treatment and the combination of pacritinib and TMZ group also demonstrated a significantly reduced CD206 mRNA level and increased TNF-α level (Fig. 6D). The overall median survival was significant increased in each treatment group compared with vehicle control (Fig. 6E). Median survival were 19 days for vehicle control, 24 days for TMZ ( p=0.024, compared with control ), 26.5 days for pacritinib ( p=0.0098, compared with control ), 32.5 day for combination of pacritinib and TMZ ( p=0.0006, compared with control, p=0.0092, compared with TMZ) (Fig. 6F).
Please kindly see our R1 Revised Figure 6 legend, Lines 317 – 328.
Figure 6 In vivo evaluation of pacritinib for treating GBM and reduction of M2 GAMs (A) Immunohistochemical staining in TMZ-resistant LN18 bearing mice showed that treatment in pacritinib group and paritinib/TMZ combination suppressed the tumorigenesis. (B) The tumor size showed that the significantly reduced tumor size in paritinib group and the combination of pacritinib and TMZ led to the most significantly reduced tumor size. NS, statistically non-significant. (C) Comparative real-time PCR analyses showed that the reduced mRNA level of STAT3, Sox2, PDCD4, and miR-21-5p while increased GFAP in pacritinib group and paritinib/TMZ combination. (D) M2 GAMs from tumor samples showed a significantly reduced CD206 (M2 marker) mRNA level (lane 3, pacritinib alone; lane 4, pacritinib/TMZ combination) while an increase in TNF-α (lane 3 and lane 4). (E) Kaplan-Meier survival curve and (F) Statistical comparisons showed that increased median overall survival in TMZ, pacritinib group and paritinib/TMZ combination. Scale lengths = 50 μm, *p<0.05; **p<0.01; ***p<0.001.< span="">
Q4: Reviewer #2: - The authors claim that miR-21-5p binds on STAT3 mRNA but nothing is described on STAT3 after Figure 4A. Can the authors add western blot for STAT3 in Figure 3C ?
A4: We thank the reviewer for this important observation. We have now added on the new Western blot for STAT3 in New Figure 3C.
Please kindly see our R1 Revised Results Section, Line 220 ~ 235.
3.3. MiR-21 is associated with GBM tumorigenic properties
Next, we examined the effects of miR-21 on GBM cells by gene silencing and overexpression techniques. First, we demonstrated that an increased miR-21 level in both U87MG and LN18 cells post co-cultured with GAM-derived exosomes (Fig. 3A). We then transfected GBM (post-incubation with GAM exosomes) with either mimic or inhibitor molecules of miR-21-5p. We found that the stemness markers Sox2, STAT3, Wnt and Nestin were all significantly increased when mimic miR-21-5p was added to both cells while the opposite occurred after the level of miR-21-5p was inhibited (Fig. 3B). In support, the results from western blots agreed with the real-time PCR results (Fig. 3C) where an increased level of miR-21-5p led to the increased expression of Sox2, STAT3, Nestin and Wnt and a decreased level of GFAP. More importantly, tumorigenic properties such as colony formation and tumor sphere formation were also positively correlated with the level of miR-21-5p. For instance, an increased miR-21-5p level by mimic molecules led to the increased number of colonies (Fig. 3D) and neurospheres (Fig. 3E) generated and the opposite was true with the decreased level of miR-21-5p with the inhibitor treatment. Furthermore, miR-21-5p mimic transfection made both U87MG and LN18 cells more resistant against TMZ whereas miR-21-5p inhibitor reversed the resistance (Fig. 3F).
Please kindly see our R1 Revised Figure 3 legend, Lines 238 – 247.
Figure 3 GAM-derived exosomes promoted GBM tumorigenesis via miR-21-5p. (A) U87MG and LN18 cells incubated with GAM-derived exosomes showed a significantly increased level of miR-21-5p. GBM tumorigenesis was associated with miR-21-5p. Increased miR-21-5p level (by mimic molecules) in GBM cells showed an increased mRNA and protein level of Sox2, Stat3, Wnt and Nestin while decreased GFAP while a decreased in miR-21-5p (inhibitor) led to the opposite phenomenon (B, C). Incubation with GAM-derived exosomes increased colony forming ability (D) and tumor sphere generating ability (E) in both U87MG and LN18 cells. (F) U87MG and LN18 cells transfected miR-21-5p mimic molecules (increased miR-21-5p level) resulted in a significantly increased TMZ resistance while reduced miR-21-59 with decreased TMZ resistance. Scale lengths = 100 μm, *p<0.05; **p<0.01; ***p<0.001.< span="">
Q5: Reviewer #2: - Many statistics were forgotten: Figures 2B, 3D, 3F, 5C, 5D, 6B. Please add them.
A4: We thank the reviewer for this important observation. We have now added on the new statistics in New Figures 2B, 3D, 3F, 5C, 5D, 6B.
Q6: Reviewer #2: - Figure 3: please define NC/M/I in the legend and there is a mistake in 3E as I and M are not on the correct panels (images are extremely blurry).
A4: We thank the reviewer for this important observation. We have now added on the new define NC/M/I in New Figure 3E.
Please kindly see our R1 Revised Results Section, Line 220 ~ 235.
3.3. MiR-21 is associated with GBM tumorigenic properties
Next, we examined the effects of miR-21 on GBM cells by gene silencing and overexpression techniques. First, we demonstrated that an increased miR-21 level in both U87MG and LN18 cells post co-cultured with GAM-derived exosomes (Fig. 3A). We then transfected GBM (post-incubation with GAM exosomes) with either mimic or inhibitor molecules of miR-21-5p. We found that the stemness markers Sox2, STAT3, Wnt and Nestin were all significantly increased when mimic miR-21-5p was added to both cells while the opposite occurred after the level of miR-21-5p was inhibited (Fig. 3B). In support, the results from western blots agreed with the real-time PCR results (Fig. 3C) where an increased level of miR-21-5p led to the increased expression of Sox2, STAT3, Nestin and Wnt and a decreased level of GFAP. More importantly, tumorigenic properties such as colony formation and tumor sphere formation were also positively correlated with the level of miR-21-5p. For instance, an increased miR-21-5p level by mimic molecules led to the increased number of colonies (Fig. 3D) and neurospheres (Fig. 3E) generated and the opposite was true with the decreased level of miR-21-5p with the inhibitor treatment. Furthermore, miR-21-5p mimic transfection made both U87MG and LN18 cells more resistant against TMZ whereas miR-21-5p inhibitor reversed the resistance (Fig. 3F).
Please kindly see our R1 Revised Figure 3 legend, Lines 238 – 247.
Figure 3 GAM-derived exosomes promoted GBM tumorigenesis via miR-21-5p. (A) U87MG and LN18 cells incubated with GAM-derived exosomes showed a significantly increased level of miR-21-5p. GBM tumorigenesis was associated with miR-21-5p. Increased miR-21-5p level (by mimic molecules) in GBM cells showed an increased mRNA and protein level of Sox2, Stat3, Wnt and Nestin while decreased GFAP while a decreased in miR-21-5p (inhibitor) led to the opposite phenomenon (B, C). Incubation with GAM-derived exosomes increased colony forming ability (D) and tumor sphere generating ability (E) in both U87MG and LN18 cells. (F) U87MG and LN18 cells transfected miR-21-5p mimic molecules (increased miR-21-5p level) resulted in a significantly increased TMZ resistance while reduced miR-21-59 with decreased TMZ resistance. Scale lengths = 100 μm, *p<0.05; **p<0.01; ***p<0.001.< span="">
Q7: Reviewer #2: - The overall quality of the figures is very poor, it is hard to correctly analyze the data and evaluate their quality: Figure 1B, graphs are blurry; Figure 3B, graph legends are impossible to read and the Western blots from 3C are too small; Figure 3E, quality of images too low, 3D is too small; Figure 4B is of very bad quality.
A3: We thank the reviewer for this important observation. We have now correctly analyzed the data and evaluate their quality in New Figure 1B, Figure 3B, Figure 3C, Figure 3D, Figure 3E and Figure 4B
Q8: Reviewer #2:- Legends of the figures are not enough detailed: authors need to precise the scale lengths, the statistical test used for the graphs.
A3: We thank the reviewer for this important observation. We have now rewritten the figure legends. Please kindly see our R1 Revised Figure 1 legend, lines 192 – 197; Revised Figure 2 legend, lines 212 – 219; Revised Figure 3 legend, lines 238 – 247; Revised Figure 4 legend, lines 264– 272; Revised Figure 5 legend, lines 289 – 298; Revised Figure 6 legend, lines 317 – 328.

Round 2
Reviewer 2 Report
I would like to thank the authors for providing most of the extra experiments in such a short time.
I still have some questions and remarks :
- I would advice the authors to cite the article "Jensen et al, Plos One 2017" as they were the first performing combo treatment with pacritinib and TMZ. I would like to thank the authors for adding the survival curves. Please give one comment on the non-significant difference between pacritinib treatment and the combo one. How can the authors explain the differences between the tumor areas in Figure 6B ?
- The authors cannot write that it is significantly increased for Stat3 in qPCR after mimic treatment, as this experiment was not done (Figure 3B). Please provide it as it is hard to evaluate by western blot.
Author Response
Answers to the comments:
Point-by-point responses to reviewer’s comments:
We would like to thank the reviewer for the thorough reading of our manuscript as well as their valuable comments. We have followed their comments closely and feel that they have further strengthened the manuscript. Below are our point-by-point responses.
Q1: Reviewer #1: I would like to thank the authors for providing most of the extra experiments in such a short time, I still have some questions and remarks. - I would advise the authors to cite the article "Jensen et al, Plos One 2017" as they were the first performing combo treatment with pacritinib and TMZ. I would like to thank the authors for adding the survival curves. Please give one comment on the non-significant difference between pacritinib treatment and the combo one. How can the authors explain the differences between the tumor areas in Figure 6B?
A1: We thank the reviewer for this important observation. We have now added on the first performing combo treatment with pacritinib and TMZ reference in discussion section.
Please kindly see our R2 Revised Discussion Section, Line 399 and 410.
Finally, we provided support for combining pacritinib and TMZ using TMZ-resistant GBM PDX mouse model. Single treatment of pacritnib was sufficient to suppress GBM growth while the combination of pacritinib and TMZ exerted the most significant inhibitory effect. Several studies have demonstrated the benefit of using STAT3 inhibitor for overcoming TMZ resistance[38,39]. Notably, one report showed that STAT3 inhibitor treatment promoted the infiltration of tumoricidal lymphocytes [40]. Another study also lend support to our results where the sequential combination of STAT3 inhibition and TMZ led to the induction of GBM apoptosis with an increased level of miR-21[41]. This is consistent with other previous study that combined treatment with pacritinib and TMZ dramatically reduced the activity of the JAK2/STAT3 pathway. This highlights the potential for pacritinib to be a useful adjuvant therapy with the standard of care TMZ. Additionally, pacritinib could be used as a salvage therapy for patients with a TMZ resistant recurrent disease, as STAT3 inhibition sensitizes TMZ resistant BTIC cultures (42).
The new reference has been added on the R2 Revised reference section, Line 582-582.
42. Jensen K.V.; Cseh O.; Aman A.; Weiss S.; Luchman H. A. The JAK2/STAT3 inhibitor pacritinib effectively inhibits patient-derived GBM brain tumor initiating cells in vitro and when used in combination with temozolomide increases survival in an orthotopic xenograft model. PLoS ONE 2017, 12(12): e0189670. https://doi.org/10.1371/journal.pone.0189670.
Q2: Reviewer #1: I would like to thank the authors for adding the survival curves. Please give one comment on the non-significant difference between pacritinib treatment and the combo one. How can the authors explain the differences between the tumor areas in Figure 6B?
A2: We thank the reviewer for this important observation. We have now added on the significant difference between pacritinib treatment and the combo one in New Figure 6F.
Please kindly see our R2 Revised Results Section, Line 298 and 314.
3.6. In vivo evaluation of pacritinib
Finally, we evaluated the potential of using pacritinib as a treatment for GBM using preclinical mouse model bearing TMZ-resistant LN18 cells (co-cultured with exosomes isolated from GAMs). Representative brain slices showed that single treatment of pancritinib suppressed the tumorigenesis of TMZ-resistant LN18 cells as compared to TMZ single treatment and vehicle control (Fig. 6A). Notably, there was no significant difference in tumor size between vehicle control and TMZ single treatment group (Fig. 6B) while the combination of pacritinib and TMZ appeared to produce the most significant inhibitory effect on tumor progression (right panel, Fig. 6B). In support, tumor samples harvested from the combination of pacritinib and TMZ showed the lowest level of STAT3, Sox2, PDCD4, and miR-21-5p while increased level of GFAP (Fig. 6C). Microglial cells isolated from the single pacritinib treatment and the combination of pacritinib and TMZ group also demonstrated a significantly reduced CD206 mRNA level and increased TNF-α level (Fig. 6D). The overall median survival was significant increased in each treatment group compared with vehicle control (Fig. 6E). Median survival were 19 days for vehicle control, 24 days for TMZ ( p=0.024, compared with control ), 26.5 days for pacritinib ( p=0.0098, compared with control ), 32.5 day for combination of pacritinib and TMZ ( p=0.0006, compared with control, p=0.0092, compared with TMZ, p=0.0219, compared with pacritinib) (Fig. 6F).
How can the authors explain the differences between the tumor areas in Figure 6B? We have now explained the differences between the tumor areas in Figure 6B due to the use with TMZ-resistant LN18 bearing mice in Revised Figure 6.
Please kindly see R2 Revised Figure 6 legend, Lines 192-197.
Figure 6 In vivo evaluation of pacritinib for treating GBM and reduction of M2 GAMs in TMZ-resistant LN18 bearing mice (A) Immunohistochemical staining in TMZ-resistant LN18 bearing mice showed that treatment in pacritinib group and paritinib/TMZ combination suppressed the tumorigenesis. (B) The tumor size showed that the significantly reduced tumor size in paritinib group and the combination of pacritinib and TMZ led to the most significantly reduced tumor size. NS, statistically non-significant. (C) Comparative real-time PCR analyses showed that the reduced mRNA level of STAT3, Sox2, PDCD4, and miR-21-5p while increased GFAP in pacritinib group and paritinib/TMZ combination (lane 1, Control; lane2, TMZ alone; lane 3, pacritinib alone; lane 4, pacritinib/TMZ combination). (D) M2 GAMs from tumor samples showed a significantly reduced CD206 (M2 marker) mRNA level (lane 3, pacritinib alone; lane 4, pacritinib/TMZ combination) while an increase in TNF-α (lane 3 and lane 4). (E) Kaplan-Meier survival curve and (F) Statistical comparisons showed that increased median overall survival in TMZ, pacritinib group and paritinib/TMZ combination. Scale lengths = 50 μm, *p<0.05; **p<0.01; ***p<0.001.< span="">
Q3: Reviewer #1: - The authors cannot write that it is significantly increased for Stat3 in qPCR after mimic treatment, as this experiment was not done (Figure 3B). Please provide it as it is hard to evaluate by western blot.
A3: We thank the reviewer for this comment. We have included a Stat3 mRNA expression in qPCR after mimic treatment in supplementary material for address the reviewer’s concern. Please kindly see our R2 New Figure 3B.
Please kindly see our R2 Revised Results Section, Line 219 and 234.
3.3. MiR-21 is associated with GBM tumorigenic properties
Next, we examined the effects of miR-21-5p on GBM cells by gene silencing and overexpression techniques. First, we demonstrated that an increased miR-21 level in both U87MG and LN18 cells post co-cultured with GAM-derived exosomes (Fig. 3A). We then transfected GBM (post-incubation with GAM exosomes) with either mimic or inhibitor molecules of miR-21-5p. We found that the stemness markers Sox2, Oct4, Wnt, STAT3 and Nestin were all significantly increased when mimic miR-21-5p was added to both cells while the opposite occurred after the level of miR-21-5p was inhibited (Fig. 3B). In support, the results from Western blots agreed with the real-time PCR results (Fig. 3C) where an increased level of miR-21-5p led to the increased expression of Sox2, Oct4, STAT3, Akt, Nestin and Wnt and a decreased level of GFAP. More importantly, tumorigenic properties such as colony formation and tumor sphere formation were also positively correlated with the level of miR-21-5p. For instance, an increased miR-21-5p level by mimic molecules led to the increased number of colonies (Fig. 3D) and neurospheres (Fig. 3E) generated and the opposite was true with the decreased level of miR-21-5p with the inhibitor treatment. Furthermore, miR-21-5p mimic transfection made both U87MG and LN18 cells more resistant against TMZ whereas miR-21-5p inhibitor reversed the resistance (Fig. 3F).
Please kindly see our R2 Revised Figure 3 legend, Lines 236 – 245.
Figure 3 GAM-derived exosomes promoted GBM tumorigenesis via miR-21-5p. (A) U87MG and LN18 cells incubated with GAM-derived exosomes showed a significantly increased level of miR-21-5p. GBM tumorigenesis was associated with miR-21-5p. Increased miR-21-5p level (by mimic molecules) in GBM cells showed an increased mRNA level of Sox2, Oct4, Wnt, STAT3 and Nestin or protein level of Sox2, Oct4, Stat3, Akt, Wnt and Nestin while decreased GFAP while a decreased in miR-21-5p (inhibitor) led to the opposite phenomenon (B and C). Incubation with GAM-derived exosomes increased colony forming ability (D) and tumor sphere generating ability (E) in both U87MG and LN18 cells. (F) U87MG and LN18 cells transfected miR-21-5p mimic molecules (increased miR-21-5p level) resulted in a significantly increased TMZ resistance while reduced miR-21-59 with decreased TMZ resistance. Scale lengths = 100 μm, *p<0.05; **p<0.01; ***p<0.001.< span="">
